# Gut microbiota profiles in feces and paired tumor and non-tumor tissues from Colorectal Cancer patients. Relationship to the Body Mass Index

**Sofía Tesolato**[1,2], **Adriana Ortega-Hernández**[2,3], **Dulcenombre Gómez-Garre**[2,3,4,5], **Paula Claver**[1], **Carmen De Juan**[1,2], **Sofía De la Serna**[2,6], **Mateo Paz**[2,7], **Inmaculada Domínguez-Serrano**[2,6], **Jana Dziakova**[2,6], **Daniel Rivera**[2,6], **Antonio Torres**[2,6], **Pilar Iniesta**[1,2]*

1 Department of Biochemistry and Molecular Biology, Faculty of Pharmacy, Complutense University, Madrid, Spain, 2 San Carlos Health Research Institute (IdISSC), Madrid, Spain, 3 Cardiovascular Risk Group and Microbiota Laboratory, San Carlos Hospital, Madrid, Spain, 4 Department of Physiology, Faculty of Medicine, Complutense University, Madrid, Spain, 5 Biomedical Research Networking Center in Cardiovascular Diseases (CIBERCV), Instituto de Salud Carlos III, Madrid, Spain, 6 Digestive Surgery Service, San Carlos Hospital, Madrid, Spain, 7 Biomedical Research Networking Center in Cancer (CIBERONC), Instituto de Salud Carlos III, Madrid, Spain

* insepi@ucm.es

**Data Availability Statement:** We have deposit the raw sequence data in a public repository: Submission ID: SUB13573780 BioProject ID:

## Abstract

Colorectal Cancer (CRC) and Obesity constitute two of the most common malignancies in the western world, and previously have been associated with intestinal microbial composition alterations. Our main aim in this study is to provide molecular data on intestinal microbiota patterns in subjects with CRC, as well as to establish possible associations with their Body Mass Index (BMI). A total of 113 samples from 45 subjects were collected and submitted to metagenomics analysis for gut microbiota. This study was performed by 16S ribosomal RNA bacterial gene amplification and sequencing using the Ion Torrent™ technology. The same dominant phyla were observed in feces and colorectal tissues, although a greater proportion of *Fusobacteriota* was found in tumor samples. Moreover, at the genus level, LEfSe analysis allowed us to detect a significant increase in *Fusobacterium* and *Streptococcus* in colorectal tissues with respect to fecal samples, with a significant preponderance of *Fusobacterium* in tumor tissues. Also, our data revealed relevant associations between gut microbiota composition and tumor location. When comparing bacterial profiles between right and left colon cancers, those from the left-sided colon showed a significant preponderance, among others, of the order *Staphylococcales*. Moreover, phyla *Firmicutes* and *Spirochaetota* were more abundant in the group of right-sided CRCs and phylum *Proteobacteria* was increased in rectal cancers. In relation to BMI of patients, we detected significant differences in beta diversity between the normal weight and the obese groups of cases. Microbiota from obese patients was significantly enriched, among others, in *Bacteroidales*. Therefore, our results are useful in the molecular characterization of CRC in obese and non-obese patients, with a clear impact on the establishment of diagnostic and prognosis of CRC.

PRJNA989099 http://www.ncbi.nlm.nih.gov/bioproject/989099.

**Funding:** The present study was supported by grant PI19/00073 from the Carlos III Institute of Health (Ministerio de Economía y Competitividad), Spain and co-funded by the European Union through the European Regional Development Fund (ERDF) "A way to make Europe". The funders had no role in study design, data collection and analysis, decision to publish, or preparation of the manuscript.

**Competing interests:** The authors have declared that no competing interests exist.

## Introduction

The impact of the intestinal microbiota on health and disease is increasingly emerging. Colorectal Cancer (CRC), one of the most common malignancies in the western world, is frequently associated with dysbiosis [1]. Epidemiologic studies have identified several lifestyle factors that affect the risk for developing CRC [2], among which obesity stands out, a condition that may be related to the composition of the intestinal microbiota. Previous works have identified *Streptococcus bovis*, enterotoxigenic *Bacteroides fragilis*, *Fusobacterium nucleatum*, *Enterococcus faecalis*, *Escherichia coli*, and *Peptostreptococcus anaerobius* as CRC candidate pathogens, and emerging studies suggest that several mechanisms, including inflammation, are closely linked to the intestinal microbiota [3]. In fact, several bacterial species have been shown to exhibit the pro-inflammatory and pro-carcinogenic properties, which could consequently have an impact on colorectal carcinogenesis. Specifically, the pro-carcinogenic properties of bacterial microbiota, such as induction of inflammation or the biosynthesis of genotoxins, interfere with cell cycle regulation [4]. Therefore, these data seem to suggest that the colon dysbiosis might contribute to CRC pathogenesis by inducing inflammation through the increase in bacterial species with known inflammatory potential [5].

Recent studies have helped to elucidate the biological differences in CRC characteristics in relation to gut microbiome and also revealed their interest in the therapy response of CRC patients [6, 7]. Moreover, considering previous works that suggest a relationship between both the composition of the intestinal microbiota and the state of obesity with the development of CRC [8], we present a study in which, as the main objective, we intend to provide molecular data on intestinal microbiota patterns in patients with CRC, as well as to establish their relationship with the Body Mass Index (BMI). Our working hypothesis is based on the consideration that the components of the tumor tissue microbiota in CRC may influence the development of the tumorigenic process and are possibly involved in the efficacy of the therapies used against cancer, as well as in the development of possible side effects. If these components can be detected in feces, it will be possible, through minimally invasive procedures, to identify biomarkers useful in the diagnosis, prognosis and establishment of personalized therapies in CRC patients. To this end, we consider it necessary to establish a study that leads to the identification of the differences in the microbiota profiles in different types of samples from patients affected by CRC (feces and paired tumor and non-tumor tissues). In addition, considering that as part of the tumor microenvironment, the gut microbiota could influence the incidence and progression of CRC by direct stimulation from bacterial bodies and metabolites [9], we correlate the intestinal microbial composition with the clinic-pathological characteristics of patients included in our protocols, such as tumor location or the BMI values of the subjects. Specifically, we carried out a study investigating intestinal microbiota in feces and paired tumor and non-tumor colorectal tissues from obese and non-obese patients affected by CRC. We analysed the components of the fecal microbiota as a possible reflection of the tissue microbiota, and we established differences in relation to clinic-pathological variables.

## Materials and methods

### Patients and samples

A total of 113 samples from 45 subjects were collected and submitted to metagenomics analysis for intestinal microbiota study. Sixty-nine samples (tumor, normal mucosa and feces) were obtained prospectively between 2021 and 2022 from 23 CRC patients who underwent potentially curative surgery. Forty-four samples (tumor and normal mucosa) had been collected retrospectively from 22 CRC patients submitted to potentially curative surgery between 2005 and 2017.

Table 1 shows age, gender and BMI values from all of the cases considered in this work, as well as data for tumor location and tumor stage for patients affected by CRC. All of the cases come from the San Carlos Hospital in Madrid (Spain). Written informed consent was obtained from patients prior to investigation. In addition, written approval to develop this study was obtained from the Clinical Research Ethics Committee of the Hospital Clínico San Carlos (C.I. 19/549-E_BC, 27/12/2019). After surgical resection, all tissue samples were instantly frozen in liquid nitrogen and stored until processed at -80˚C in the Biobank of the Health Research Institute of San Carlos Hospital (IdISSC) in Madrid, using Tissue-Tek OCT as a freezing medium. Therefore, all tissues from CRC patients were obtained from the San Carlos Hospital Biobank (B.0000725) belonging to the San Carlos Health Research Institute (IdISSC), which is part of the national network of Biobanks, project PT2020/00074 subsidized by the Carlos III Institute of Health (ISCIII) and co-funded by the European Union through the European Regional Development Fund (ERDF).

Cryostat sectioned, H&E stained samples from each tumour block were examined microscopically by two independent pathologists to confirm the presence of $\geq$ 80% tumor cells. Paired normal tissues from the same patients were obtained and microscopically confirmed. All CRCs were staged according to the NCCN guidelines (National Comprehensive Cancer Network) v 2.2022 [10]. Patients were classified according to their BMI values following the criteria of the World Health Organization (WHO). Thus, patients with BMI $\leq$ 24.9 Kg/m$^2$ were considered to have normal weight (n = 15); patients with BMI $\geq$ 25 kg/m$^2$ and $\leq$ 29.9 kg/m$^2$, as overweight (n = 13); and the ones with BMI $\geq$ 30 kg/m$^2$ were defined as with obesity (n = 17). Cases were collected independently from gender, age of the patient or tumor stage. Moreover, no CRC patient had received chemo- or radiotherapy before surgery and inclusion in the study. Patients who had previously undergone digestive surgery, those affected by inflammatory diseases and those who had undergone antibiotic treatment one month before surgery were excluded.

## DNA preparation and sequencing

Tissue-Tek OCT from tissue samples was removed prior to DNA extraction by washing with 1ml PBS 1x and centrifuging at 3000 rpm during 15 minutes at room temperature. Next, total

**Table 1. Clinic-pathological characteristics of subjects with Colorectal Cancer.**

| Variable | Colorectal Cancer (N = 45 patients) |
|---|---|
| **Mean age ± standard error, years** | 74.27 ± 1.52 |
| **Gender, N (%)** | |
| Male | 27 (60) |
| Female | 18 (40) |
| **BMI group, N (value, mean ± standard error)** | |
| Normal weight (BMI $\leq$ 24.9 Kg/m$^2$) | 15 (23.39 ± 0.26) |
| Overweight (BMI $\geq$ 25 kg/m$^2$ and $\leq$ 29.9 kg/m$^2$) | 13 (27.09 ± 0.29) |
| Obesity (BMI $\geq$ 30 kg/m$^2$) | 17 (32.45 ± 0.55) |
| **Tumor location, N (%)** | |
| Right colon | 16 (35.56) |
| Left colon | 16 (35.56) |
| Rectum | 13 (28.88) |
| **TNM stage, N (%)** | |
| I-II | 26 (57.78) |
| III-IV | 19 (42.22) |

DNA from prospective colorectal samples was obtained using the QIAamp® DNA Mini Kit (Qiagen, Hilden, Germany). DNA from feces was extracted using the QIAamp® Fast DNA Stool Mini Kit (Qiagen, Hilden, Germany). The concentration of the DNA extracts was determined with the Invitrogen™ Qubit™ 3 Fluorometer using the dsDNA HS (High Sensitivity) Assay (Thermo Fisher Scientific, Madrid, Spain). DNA from colorectal retrospective samples had been obtained according to the Blin and Stafford procedure [11].

Microbiota analysis was performed by 16S ribosomal RNA (rRNA) bacterial gene amplification and sequencing, using the Ion Torrent™ sequencing technology and the reagents from Life Technologies S.A. (a part of Thermo Fisher Scientific), as previously described [12]. Briefly, for each sample, a starting input of 5ng of DNA was used to run 2 parallel polymerase chain reactions (PCRs), which amplified seven hypervariable regions of the gene (V2, V4, V8 and V3, V6-7, V9, respectively) with two different sets of primers, using the Ion 16S™ Metagenomics Kit. The PCR conditions followed the manufacturer's protocol, and both PCR reactions were set to 25 cycles. The amplicons obtained from both reactions were then quantified, combined for each sample in equal volumes and purified using the Agencourt® AMPure® XP Reagent beads (Beckman Coulter, Madrid, Spain). Next, barcoded libraries were prepared using the Ion Plus Fragment Library Kit and the Ion Xpress™ Barcode Adapters. The libraries were then quantified with the Qubit™ 3 Fluorometer, set to 22pM, and then combined in the same tube. An emulsion PCR was performed using the Ion OneTouch™ 2 System together with the Ion 520™ & 530™ Kit–OT2. The template-positive Ion Sphere Particles (template-positive ISPs) were then collected, washed and enriched using the Ion OneTouch™ ES Instrument OT2, and finally loaded onto an Ion 530™ Chip and sequenced using the Ion S5™ System.

## Statistical analysis

Statistical Analysis was performed using the Quantitative Insight Into Microbial Ecology 2 (QIIME2) pipeline. QIIME2 pre-processing was established through the following steps. First, sequences were pre-processed using Torrent suite (v5.10 Thermo Fisher Scientific). Using our custom Python pipeline, primers were removed and sequences were trimmed to 150 bp. QIIME 2 was used to perform de-replication, remove singletons and filtering chimeras. Rarefaction was performed. OTUs abundances from different 16S regions were summed to create a single taxonomy table. Relative OTUs abundances were computed as proportions (%) based on the total number of features per sample. Alfa diversity was performed based on a rarefied OTUs profile. The SILVA 138 SSU Ref NR 99 identity database was used for closed-reference OTUs picking. Alpha diversity was assessed conducting parametric tests, in the case of data normally distributed (t-test) or non-parametric tests, in the case of data with a non-normal distribution (Kruskal Wallis test). Five metrics were calculated: observed OTUs, Chao1 richness estimate, Shannon diversity index, Pielou's evenness index and Simpson's diversity index. For beta diversity analysis, Permutation-based multivariate analysis of variance (PERMANOVA), Analysis of similarities (ANOSIM) and PERMDISP2 tests were performed, comparing Jaccard and Bray-Curtis similarity indexes, with distances between groups plotted using principal coordinate analysis (PCoA).

A linear discriminant analysis (LDA) effect size (LEfSe) was used for measuring the relative abundances of taxa in the analysed groups. In order to facilitate the understanding of these data, we clarify that the level of LDA analyses that we showed depended on the number of taxonomic differences present in each particular comparison. Even though we are aware that the differences in bacterial genera may be more specific, when the differences were more abundant (e.g. in the comparison between feces and tumor or non-tumor tissue), we found more interesting to focus on the differences at the higher taxonomic levels (phyla, orders. . .) as they were

more manageable and less confusing than the ones at lower taxonomic levels (LDA at the genus level were very big and difficult to include in a figure). However, when the differences were slighter (e.g. between tumor and non-tumor colorectal tissues), these did not reach the phyla or order levels, so in this case we included the LDA at the genus level with the purpose of refining the comparison.

LEfSe consists of the application of a Kruskal–Wallis test to identify taxa with significantly different relative abundances, followed by an LDA to determine an effect size of each taxon. Taxa with an LDA score (log10) > 2 with a P < 0.05 were considered statistically significant.

As quality control, the sequencing depth distribution had a median frequency of 234.647 ranging [27.182–448.128] reads per sample.

## Results

### Gut microbiota alpha diversity comparison between feces and non-tumor samples from CRC patients

Considering CRC samples, firstly we analysed differences in gut microbiota between feces and non-tumor tissues from patients. Alpha diversity comparison between feces and non-tumor colorectal tissues reported significant differences in both the Observed OTUs (P < 0.001) and the Chao1 index (P = 0.015), and bordering significance in the Shannon index (P = 0.058) (Fig 1). The three metrics were higher in non-tumor tissues, indicating a greater microbial diversity in this type of sample (mean value ± standard error: Observed OTUs, 2300 ± 69 in feces and 2786 ± 85 in non-tumor tissues; Chao1 index, 4683 ± 164 in feces and 5407 ± 188 in non-tumor tissues; Shannon index, 8 ± 0.13 in feces and 8.23 ± 0.15 in non-tumor tissues).

### Gut microbiota beta diversity comparison between feces and non-tumor samples from CRC patients

The differences in gut microbial composition between feces and non-tumor colorectal tissue samples are shown in Fig 2. The two types of samples showed that the dominant bacterial phyla in abundance were *Firmicutes*, *Bacteroidota*, *Proteobacteria* and *Actinobacteriota* (Fig 2a). PERMANOVA test for beta diversity showed significant differences between both groups, both in Bray-Curtis (P = 0.001) and Jaccard (P = 0.001) indexes (Fig 2b). LEfSe analysis at the bacterial phylum level showed significant differences between feces and non-tumor tissue

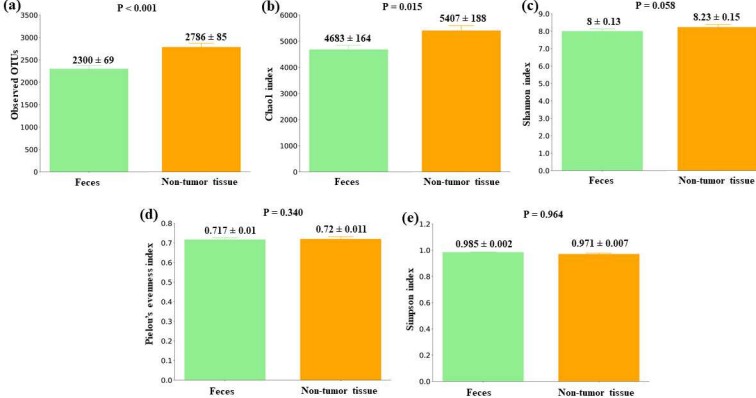

**Fig 1. Alpha diversity comparison between feces and non-tumor colorectal tissues from CRC patients.** (a) Observed OTUs (b) Chao1 index (c) Shannon index (d) Pielou's evenness index (e) Simpson index. Mean values ± standard error and P values are indicated.

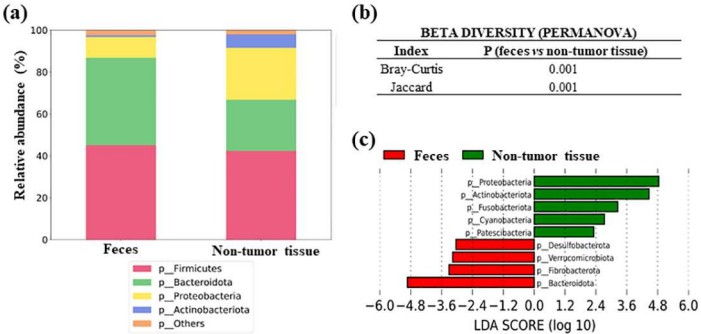

**Fig 2. Microbiota composition in feces and non-tumor colorectal tissues from CRC patients.** (a) Comparison of the relative abundance of the main bacterial phyla (b) Permanova test for beta diversity (c) LEfSe comparing bacterial phyla between both samples.

samples. The non-tumor tissue microbiota was characterized by an overabundance of phyla *Proteobacteria* (P = 0.002), *Actinobacteriota* (P < 0.001), *Fusobacteriota* (P = 0.007), *Cyanobacteria* (P = 0.034), and *Patescibacteria* (P < 0.001), whereas the main phylum *Bacteroidota* (P < 0.001) and other less abundant phyla (*Desulfobacterota*, P = 0.019; *Verrucomicrobiota* P < 0.001; and *Fibrobacterota*, P < 0.001) were increased in fecal samples (Fig 2c).

## Gut microbiota alpha diversity comparison between feces and tumor samples from CRC patients

As it can be observed in Fig 3, in the case of tumor tissues, alpha diversity comparison to fecal samples showed differences in the Observed OTUs (P = 0.024) and in the Chao1 index (bordering significance, P = 0.079), which were higher in the tumor tissue samples (mean value ± standard error: Observed OTUs, 2650 ± 81 in feces and 3036 ± 111 in tumor tissues; Chao1 index, 5254 ± 186 in feces and 5734 ± 240 in tumor tissues).

## Gut microbiota beta diversity comparison between feces and tumor samples from CRC patients

The differences in microbial composition between feces and tumor colorectal samples are shown in Fig 4. The same dominant phyla (*Firmicutes*, *Bacteroidota*, *Proteobacteria* and

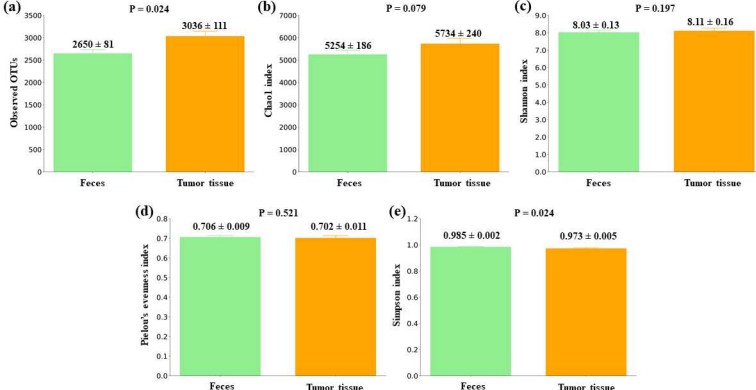

**Fig 3. Alpha diversity comparison between feces and tumor colorectal tissues from CRC patients.** (a) Observed OTUs (b) Chao1 index (c) Shannon index (d) Pielou's evenness index (e) Simpson index. Mean values ± standard error and P values are indicated.

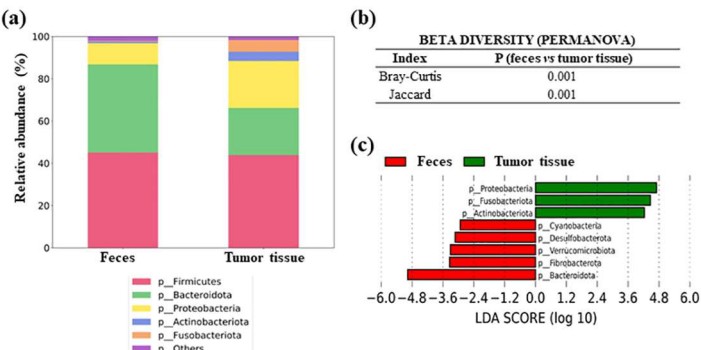

**Fig 4. Microbiota composition in feces and tumor colorectal tissues from CRC patients.** (a) Comparison of the relative abundance of the main bacterial phyla (b) Permanova test for beta diversity (c) LEfSe comparing bacterial phyla between both samples.

*Actinobacteriota*) were observed in tumor tissue samples, although a greater proportion of *Fusobacteriota* was also observed (Fig 4a). These phyla were also detected in feces from patients. PERMANOVA test for beta diversity showed significant differences between groups, both in Bray-Curtis and Jaccard indexes (P = 0.001 for both) (Fig 4b).

LEfSe analysis at the bacterial phylum level revealed some common differences with the feces-non tumor tissue comparison: four phyla were also increased in feces (*Bacteroidota*, P < 0.001; *Fibrobacterota*, P < 0.001; *Verrucomicrobiota*, P < 0.001 and *Desulfobacterota*, P = 0.010, and the same phyla *Proteobacteria* (P = 0.010), *Actinobacteriota* (P < 0.001) and *Fusobacteriota* (P < 0.001) increased in tissue samples. However, phylum *Cyanobacteria* was increased in feces when compared to tumor tissue (P = 0.040) (Fig 4c). Notably, when performing LEfSe analysis at the genus level, we found that two genera (*Fusobacterium* and *Streptococcus*) were significantly increased in tumor and non-tumor colorectal samples with respect to fecal samples (LDA in tumor colorectal samples: P < 0.001 for *Fusobacterium*, and P = 0.002 for *Streptococcus*; LDA in non-tumor colorectal samples: P = 0.014 for both *Fusobacterium* and *Streptococcus*).

## Gut microbiota comparison between tumor and non-tumor samples from CRC patients

We investigated possible microbiota differences between tumor and non-tumor tissues (Fig 5). No significant differences were found in any of the alpha and beta diversity tests. However,

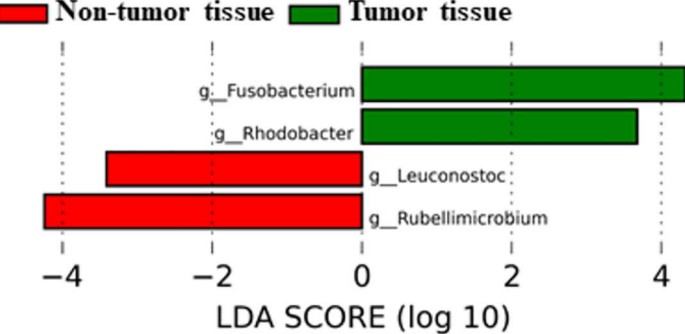

**Fig 5. LEfSe in bacterial genus between tumor and non-tumor colorectal tissues from CRC patients.**

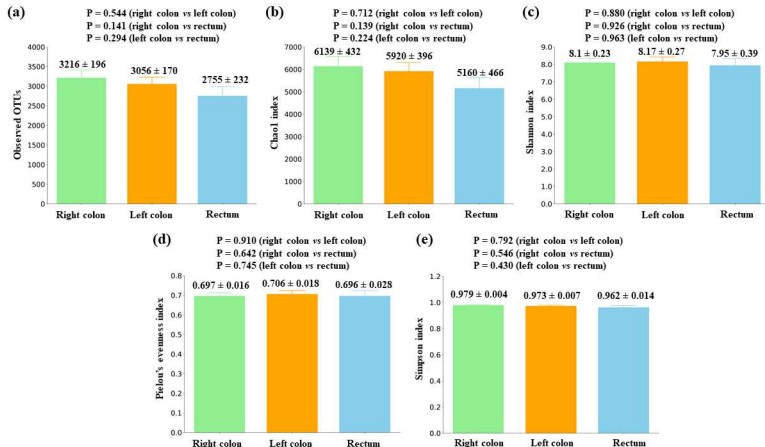

**Fig 6. Alpha diversity comparison between colorectal tumor tissues with different locations.** (a) Observed OTUs (b) Chao1 index (c) Shannon index (d) Pielou's evenness index (e) Simpson index. Mean values ± standard error and P values are indicated.

LEfSe analysis at the genus level revealed that *Fusobacterium* (P = 0.034) and *Rhodobacter* (P = 0.046) were increased in tumor tissues, and genera *Rubellimicrobium* (P = 0.012) and *Leuconostoc* (P = 0.033) in non-tumor tissues.

## Gut microbiota differences according to tumor location in CRC patients

Considering tumor characteristics and other clinical variables, we detected important differences in relation to tumor location. Alpha diversity comparison between colorectal tumor tissue samples from different locations did not report any significant differences (Fig 6).

With respect to microbiota composition, the analysis of PERMANOVA test for beta diversity showed significant differences between right and left-sided colorectal tumors (P = 0.041 for Bray-Curtis index and bordering significance P = 0.065 for Jaccard index) (Fig 7a), as well as between right colon and rectum tumors (P < 0.001 for Bray-Curtis and Jaccard index) (Fig 7b).

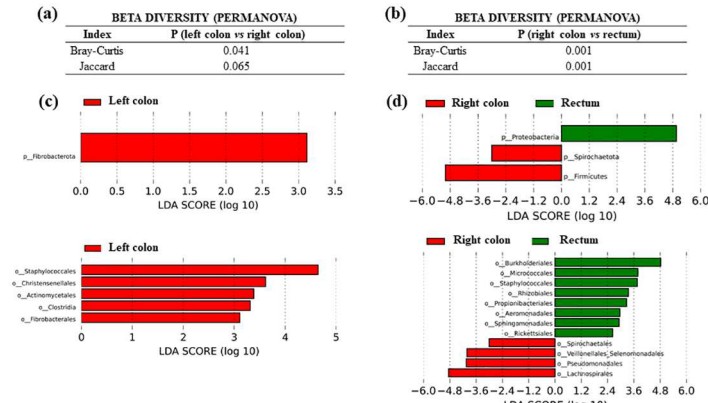

**Fig 7. Comparison in microbiota composition between colorectal tumor tissues with different locations from CRC patients.** (a) Permanova test for beta diversity between left- and right-sided colon tumors. (b) Permanova test for beta diversity between right-sided colon and rectum tumors. (c) LEfSe comparison between left- and right-sided colon tumors. (d) LEfSe comparison between right-sided colon and rectum tumors.

LEfSe analysis revealed significant differences in bacterial populations between right and left colon cancers (Fig 7c). At the bacterial phylum level, left-sided colon tumor microbiota was significantly enriched in *Fibrobacterota* (P = 0.022). A deeper analysis at the bacterial order level revealed that left-sided colon tumor microbiota was characterized by a preponderance of *Staphylococcales* (P < 0.001, from phylum *Firmicutes*), *Christensenellales* (P = 0.008, from phylum *Firmicutes*), *Actinomycetales* (P = 0.004, from phylum *Actinobacteriota*), *Clostridia* (P = 0.013, from phylum Firmicutes) and *Fibrobacterales* (P = 0.022, from phylum *Fibrobacterota*). The relative abundance of bacterial taxa also differed significantly when LEfSe was performed between right colon and rectum tumor samples (Fig 7d). Our results indicated that phyla *Firmicutes* and *Spirochaetota* were more abundant in the group of right-sided CRCs (P = 0.007 and P = 0.018, respectively), whereas phylum *Proteobacteria* was increased in rectal cancers (P = 0.016). At the bacterial order level, three orders belonging to the *Firmicutes* phylum (*Lachnospirales*, P = 0.003; *Veillonellales*, P = 0.004; and *Selenomonadales*, P = 0.004), one order belonging to the *Proteobacteria* phylum (*Pseudomonadales*, P = 0.029), and one order belonging to the *Spirochaetota* phylum (*Spirochaetales*, P = 0.018), were more abundant in righ-sided CRCs. In contrast, rectal tumors showed an increase in one order belonging to the *Firmicutes* phylum (*Staphylococcales*, P < 0.001), two orders from the *Actinobacteriota* phylum (*Micrococcales*, P = 0.011; and *Propionibacteriales*, P = 0.018), and five orders from the *Proteobacteria* phylum (*Burkholderiales*, P = 0.008; *Rhizobiales*, P = 0.029; *Aeromonadales*, P = 0.041; *Sphingomonadales*, P = 0.018; and *Rickettsiales*, P = 0.040).

With respect to the comparison between left colon and rectal tumor microbiota, there were no significant differences in any of the beta diversity tests performed (PERMANOVA test: P = 0.270 for Bray-Curtis index and P = 0.277 for Jaccard index; ANOSIM test: P = 0.444 for Bray-Curtis index and P = 0.354 for Jaccard index; PERMDISP test: P = 0.332 for Bray-Curtis index and P = 0.538 for Jaccard index). At the taxonomic level, there were only slight differences between both locations, which had been previously seen between right colon and rectal samples: phylum *Firmicutes* was again increased in left colon tumors, P = 0.016, and order *Aeromonadales* in rectal tumors, P = 0.039.

## Gut microbiota differences in subjects with CRC considering BMI values

Given the influence of obesity on the risk of developing CRC, we analysed possible differences in the gut microbiota profiles in tissues from CRC patients, in relation to their BMI values. These analyses were performed jointly in all of the colorectal tissues included in the present study (tumor and non-tumor tissues), since no significant differences had been found in any of the alpha and beta diversity tests between both types of tissues.

No significant differences in the alpha diversity metrics were obtained between the BMI groups (Fig 8).

Regarding microbial composition, we detected significant differences in beta diversity between the normal weight and the obese groups of cases (Bray-Curtis index in PERMANOVA test, P = 0.017 and Jaccard index bordering significance, P = 0.063) (Fig 9a). When performing LEfSe analysis at the bacterial order level, the tissue microbiota from normal weight CRC patients was characterized by a preponderance of the following orders: *Christensenellales* (P = 0.022) and *Clostridiales* (P = 0.013), from the phylum *Firmicutes*; and *Campylobacterales* (P = 0.002), from the phylum *Campylobacterota*. In contrast, the microbiota of obese patients was significantly enriched with orders *Bacteroidales* (P = 0.005), phylum *Bacteroidota*; *Desulfovibrionales* (P = 0.034), phylum *Desulfobacterota*; and *Sphingobacteriales* (P = 0.032), phylum *Bacteroidota*. (Fig 9b).

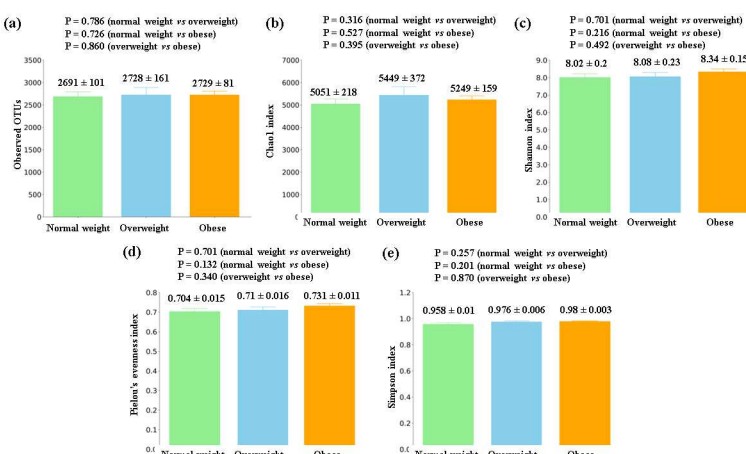

**Fig 8. Alpha diversity comparison between colorectal tissues from CRC patients with different BMI.** (a) Observed OTUs (b) Chao1 index (c) Shannon index (d) Pielou's evenness index (e) Simpson index. Mean values ± standard error and P values are indicated.

When performing microbiota comparison between colorectal tissues from normal weight and overweight CRC patients, there were significant differences in beta diversity between both groups for PERMANOVA test (P = 0.001 for Bray-Curtis and Jaccard indexes) and for ANO-SIM test (P = 0.002 for Bray-Curtis index and P = 0.001 for Jaccard index). LEfSe analysis in bacterial order revealed some common differences with the normal weight-obese comparison: order *Christensenellales* was also increased in the tissue microbiota from normal weight patients (P = 0.006), whereas order *Sphingobacteriales* was increased in the overweight group (P = 0.034).

Comparisons in the microbiota composition between feces from normal weight and obese CRC patients did not report any statistically relevant differences in alpha and beta diversity tests, although the Jaccard index in ANOSIM beta diversity test was bordering statistical significance (P = 0.098). However, genus *Odoribacter*, from the phylum *Bacteroidota*, was significantly increased in feces from obese CRC patients (P = 0.019) (Fig 10).

When feces from normal weight CRC patients were compared to the ones from overweight patients, beta diversity was significantly different in two of the three tests performed (PERMA-NOVA test, P = 0.024 for Bray-Curtis index and P = 0.056 for Jaccard index; and ANOSIM test, P = 0.005 for Bray-Curtis index and P = 0.017 for Jaccard index). Interestingly, LEfSe analysis at the genus level reported that genus *Odoribacter* was also increased in the feces from the overweight group (P = 0.024).

As a summary of the results obtained in this work investigating CRC samples, we highlight the following items:

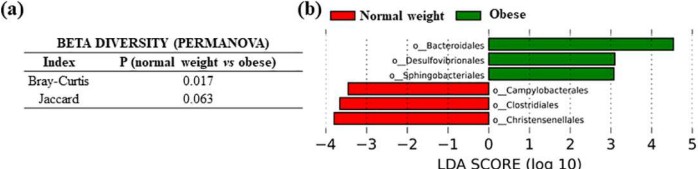

**Fig 9. Microbiota comparison in colorectal tissue samples from CRC patients (normal weight *versus* obese).** (a) Permanova test for beta diversity (b) LEfSe in bacterial order.

**Fig 10. Microbiota comparison in feces from CRC patients (normal weight *versus* obese).** (a) Anosim test for beta diversity (b) LEfSe in bacterial order.

- Alpha diversity analysis indicated that both tumor and non-tumor tissues showed a greater microbial diversity than feces, and significant differences in beta diversity.

- From LEfSE analysis, we observed that two genera (*Fusobacterium* and *Streptococcus*) were significantly increased in tumor and non-tumor colorectal tissues with respect to fecal samples. *Fusobacterium* also showed a preponderance in tumor tissues.

- Significant differences in microbiota composition were detected between tumors from the right and the left-sided colon, as well as between right colon and rectum tumors.

- Significant differences in beta diversity were observed in relation to the BMI values of CRC patients. Specifically, relevant differences were detected between colorectal tissues of normal weight and obese CRC subjects. The latter showed a significant enrichment in orders of the phylum *Bacteroidota*. In addition, the feces of obese CRC patients reflected the preponderance of this bacterial phylum (genus *Odoribacter*).

## Discussion

An increasing number of studies have revealed that the progression of CRC is related to the gut microbiota composition. Under normal conditions, the gut microbiota acts as a barrier to other pathogens or infections in the intestine and modulates inflammation by affecting the host immune system [13]. Our results in CRC patients, investigating microbiota in feces and paired non-tumor and tumor colorectal tissues, indicated significant differences for beta diversity between feces and both non-tumor and tumor samples. Also, we found no significant differences when we analysed beta diversity between both types of tissues. Moreover, LEfSe analysis at the genus level allowed us to detect two genera which contain bacterial species that had been previously associated with CRC, *Fusobacterium* and *Streptococcus*. Both genera were significantly increased in tumor and non-tumor colorectal samples with respect to fecal samples. Therefore, these data seem to indicate that microbiota analysis in feces can be considered only a representation of the colorectal tissue microbiota.

In the bibliography, there are very few studies that compare microbiota results between feces and colorectal tissues from patients [14]. The relevance of intestinal bacteria and their metabolites in CRC has been a research topic in the last years. Stool, blood, tissue, and bowel fluid are the main sample sources investigated in the search for biomarkers [14]. The fecal microbiome has been more intensively studied for the early detection of CRC, and increasing evidence has supported correlations between CRCs and fecal microbiota dysbiosis [15–17]. However, fewer studies have been published that provide convincing results for the characterization of the intestinal tissue microbiota. Metagenomic studies in human CRC associated microbiome signatures with the colorectal adenoma-carcinoma sequence, establishing a critical function for the intestinal microbiota in promoting tumorigenesis [1]. Mo et al reported a high correlation between colorectal carcinoma adjacent tissue microbiomes and their on-site counterparts, suggesting that microbial feature variations of cancerous lesion-adjacent tissues might

help to reveal the microbial etiology of colonic cancer [18]. These results seem to coincide with those reported by our group in the present work. In fact, we present comparisons between microbiota composition in tumor and non-tumor tissues, finding no significant differences in beta diversity between both groups of samples. Previously, slightly lower taxonomic diversity was observed in the tumor compared to the non-tumor tissues [19]. Our data (LEfSe analysis) revealed a significant preponderance of the *Fusobacterium* genus in tumor tissues. Recent investigations have suggested an association between certain members of the *Fusobacterium* genus, especially *F. nucleatum*, and the progression of advanced colorectal carcinoma [20]. The current studies show that *F. nucleatum* can actively participate in the occurrence and metastasis of CRC by affecting a variety of mechanisms including, among others, immune regulation [21].

In relation to clinical variables of CRCs, we detected relevant associations between gut microbiota composition and tumor location. We also detected relevant differences in the beta diversity analysis when tumor location was considered. Moreover, LEfSe analysis indicated significant differences in bacterial populations in relation to CRCs location. Tumors from the left-sided colon showed a preponderance of the order *Staphylococcales*, when compared to the right-sided colon tumors. Therefore, our data seem to indicate that microbiota composition differs depending on CRC anatomical location. These results could be related to the distinctive molecular characteristics of the right and left CRCs, as well as the rectum cancers. Our results would be in line with other works, supporting that the understanding of the microbiota profiles at different sites of colon and rectum may help to better define the molecular subtypes of CRC [22]. Furthermore, it is interesting to consider that previously it has been reported that most of right-sided CRCs are associated with a dense bacterial biofilm and that the organization of the mucosa-associated microbial community is an important factor in CRC pathogenesis, particularly in the proximal colon [23]. The authors of this paper hypothesized that the risk of developing CRC is more than fivefold higher in the patients with biofilms compared with those without biofilms.

The results found in our study allow us to conclude that, although there are significant differences between the intestinal microbiota patterns at the tissue and fecal levels, the stool analysis is valid for detecting important microorganisms associated with the development of CRC. Therefore, the data included in this manuscript support, in agreement with other authors, that feces constitute a valid material in the study of the intestinal microbiota [14]. In addition, the assessment of a patient's microbial composition represents a key element of the future precision medicine approach [24].

Finally, we present the results obtained investigating the intestinal microbiota profiles in relation to the BMI values of the CRC individuals considered. In recent years, increasing evidence linking obesity to the gut microbiota has been reported. However, the complex interactions among genetics, environment, the gut microbiota, and obesity remain poorly understood [25]. Chronic inflammation is one of the characteristics of metabolic disorders such as obesity [26]. Inflammation increases the probability of developing cancer and several bacterial species have been shown to exhibit the pro-inflammatory and pro-carcinogenic properties, which could consequently have an impact on colorectal carcinogenesis [4]. Moreover, obesity and high fat diet (HFD) have been linked to changes in gut microbiota composition, with a decrease in bacteria producing beneficial short chain fatty acids (SCFAs), mainly butyrate, and an increase in lipopolysaccharide (LPS)-producing pathogens, which along with an increased intestinal permeability leads to higher circulating LPS, endotoxemia and low-grade inflammation [27, 28]. Evidence shows that these disorders are characterized by the gut microbiota and its metabolites crossing the intestinal barrier, affecting various metabolic organs, such as the liver and adipose tissue, and leading to chronic inflammation [29].

Considering results from our study, we observed significant differences in the beta diversity between the colorectal tissues from the normal weight and the obese CRC groups of patients,

as well as important differences in bacterial taxa. Particularly, colorectal tissue from obese CRC patients showed an increase in order *Desulfovibrionales*, which contains several sulfidogenic bacteria, such as *Bilophila wadsworthia*, previously related to increased BMI values, as well as with the development of CRC via the production of genotoxic hydrogen sulfide [30–32]. Moreover, *B. wadsworthia* has been related to increased LPS biosynthesis and translocation [33]. Order *Bacteroidales*, containing the previously CRC-associated *Bacteroides fragilis* [3], was also found to be increased in this group of patients. On the other hand, colorectal mucosa from normal weight CRC patients was enriched in order *Christensenellales*, strongly related to health and a normal BMI in several studies [34].

Remarkably, this difference was noticeable both when normal weight patients were compared to obese patients and when they were compared to overweight patients, thus reinforcing the association of this type of bacteria with a healthy body mass.

Analysis in feces from CRC patients in relation to their BMI indicated that genus *Odoribacter* was increased in feces from patients with excess body mass (either overweight or obese) with respect to normal weight patients, suggesting an influence of BMI. The relationship of *Odoribacter* with CRC is controversial. Although this bacterial genus is detected at increased levels in patients with CRC, some studies attribute a protective function to it due to the production of cytokines and the induction of immune cells [35], while others advocate for a pro-tumorigenic role due to the production of genotoxic substances [36]. Furthermore, a decrease in *Odoribacter* levels has been associated with obesity events [37]. Therefore, the increase in *Odoribacter* levels in CRC patients with obesity could be exclusively due to the tumor events.

As the main limitation of this work, we consider the heterogeneity in age and sex of the populations under study, which could influence the intestinal microbial composition. Another limitation derives from the small number of cases analysed, which complicates the statistical adjustment for multiple comparisons.

## Conclusions

We consider that the results from this work are useful in the molecular characterization of CRC in obese and non-obese patients, which may have an impact on the establishment of diagnostic, prognosis and therapy of CRC. More specifically, our data support the following conclusions. Firstly, CRC microbiota composition differs according to the anatomical location of tumors, which could be related to the distinctive molecular characteristics of the right- and left-sided colon and rectum cancers. Secondly, in colorectal tissues we detected significant differences in beta diversity indexes in relation to the BMI values of CRC patients. Finally, analysis of feces and colorectal tissues from CRC patients shows less microbial diversity in the first type of sample, although markers associated with CRC and obesity are detected. Among them, we highlight the genus *Fusobacterium* and genera from the phylum *Bacteroidota*, respectively.

## Author Contributions

**Conceptualization:** Dulcenombre Gómez-Garre, Antonio Torres, Pilar Iniesta.

**Formal analysis:** Sofía Tesolato, Adriana Ortega-Hernández, Dulcenombre Gómez-Garre, Carmen De Juan, Mateo Paz, Antonio Torres, Pilar Iniesta.

**Funding acquisition:** Antonio Torres, Pilar Iniesta.

**Investigation:** Sofía Tesolato, Adriana Ortega-Hernández, Dulcenombre Gómez-Garre, Paula Claver, Carmen De Juan, Mateo Paz, Pilar Iniesta.

**Methodology:** Sofía Tesolato, Adriana Ortega-Hernández, Dulcenombre Gómez-Garre, Sofía De la Serna, Inmaculada Domínguez-Serrano, Jana Dziakova, Daniel Rivera, Pilar Iniesta.

**Project administration:** Antonio Torres, Pilar Iniesta.

**Supervision:** Dulcenombre Gómez-Garre, Sofía De la Serna, Pilar Iniesta.

**Validation:** Sofía Tesolato, Dulcenombre Gómez-Garre, Sofía De la Serna, Pilar Iniesta.

**Writing – original draft:** Sofía Tesolato, Paula Claver, Antonio Torres, Pilar Iniesta.

**Writing – review & editing:** Sofía Tesolato, Sofía De la Serna, Jana Dziakova.

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
