## [Decision Letter · Decision Letter 0]

28 Mar 2023

PONE-D-23-02427Gut microbiota profiles in feces and paired tumor and non-tumor tissues from Colorectal Cancer patients. Relationship to the Body Mass IndexPLOS ONE

Dear Dr. Iniesta,

Thank you for submitting your manuscript to PLOS ONE. After careful consideration, we feel that it has merit but does not fully meet PLOS ONE’s publication criteria as it currently stands. Therefore, we invite you to submit a revised version of the manuscript that addresses the points raised during the review process. Both reviewers found the manuscript of interest but expressed significant concern about the analyses shown in this manuscript, which could be difficult to follow and give the impression of “cherry picking.” In particular, Reviewer #1 presents valid criticism of the small sample size of subgroups, in particular the control group. I agree that the non-CRC control group is too small (n=9) and demographically different to be convincingly used for case vs. control comparisons. I also agree that the separate analysis of the prospective and retrospective CRC subsets leads to confusion and that these should be combined into an analysis of a single larger cohort with greater power if possible.

Please revise the analyses in the manuscript to remove the non-CRC group, combine the retrospective and prospective cohorts for analyses of the CRC tissue microbiome, and address the questions raised by Reviewer #2.

We look forward to receiving your revised manuscript.

Kind regards,

Jonathan Jacobs

Academic Editor

PLOS ONE

https://journals.plos.org/plosone/s/file?id=ba62/PLOSOne_formatting_sample_title_authors_affiliation’s.pdf

“The present study was supported by grant PI19/00073 from the Carlos III Institute of Health (Ministerio de Economía y Competitividad), Spain and co-funded by the European Union through the European Regional Development Fund (ERDF) "A way to make Europe".”

Additional Editor Comments:

Both reviewers found the manuscript of interest but expressed significant concern about the analyses shown in this manuscript, which could be difficult to follow and give the impression of “cherry picking.” In particular, Reviewer #1 presents valid criticism of the small sample size of subgroups, in particular the control group. I agree that the non-CRC control group is too small (n=9) and demographically different to be convincingly used for case vs. control comparisons. I also agree that the separate analysis of the prospective and retrospective CRC subsets leads to confusion and that these should be combined into an analysis of a single larger cohort with greater power if possible.

Please revise the analyses in the manuscript to remove the non-CRC group, combine the retrospective and prospective cohorts for analyses of the CRC tissue microbiome (possibly with retrospective vs. prospective as a covariate as suggested by Reviewer #1), and address the questions raised by Reviewer #2.

Reviewers' comments:

Reviewer's Responses to Questions

**Comments to the Author**

1. Is the manuscript technically sound, and do the data support the conclusions?

Reviewer #1: No

Reviewer #2: Yes

2. Has the statistical analysis been performed appropriately and rigorously? 

Reviewer #1: N/A

Reviewer #2: Yes

3. Have the authors made all data underlying the findings in their manuscript fully available?

Reviewer #1: No

Reviewer #2: Yes

4. Is the manuscript presented in an intelligible fashion and written in standard English?

Reviewer #1: Yes

Reviewer #2: Yes

5. Review Comments to the Author

Reviewer #1: The authors conducted 16S rRNA microbiome profiling (not metagenomics) in 45 colorectal cancer (CRC) cases and 9 healthy controls. They conducted multiple comparisons (with no statistical adjustments for multiple comparisons) including, in cases, tumor vs non-tumor mucosal tissue (no differences), right- versus left-sided tumors (significant alpha and beta diversity differences), and obese vs non-obese feces (possible beta diversity differences). They also compared feces between cases and controls (no differences).

Overall, the project succeeded as a technical demonstration. However, each of the groups is too small to have confidence in the results (whether "significant" or null). The Results Figures (11 in total) are a good example of cherry picking.

The limitations sentence (L 494-495) is lacking. The control group is without value -- too small, too young, and too female.

Given the technical success, continuing the work would be appropriate, but giving much more thought into the objectives (e.g., either only study CRC cases or be much more thoughtful and proactive in selecting and recruiting appropriate controls). For the current "Results" I suggest splitting it into 2-3 Letters to the Editor (completely dropping the controls). In addition, they could probably drop the separation of the prospective and retrospective CRC cases; simply present a brief comparisons, then statistically adjust for prospective / retrospective.

Reviewer #2: This paper by Sofía Tesolato, et al. is very interesting paper regarding to gut microbiota profiles in feces and paired tumor and non-tumor tissues from Colorectal Cancer patients. These results and findings provide the important evidence to researchers in this field.

However, several points should be concerned.

1. There are many LDA analysis in the results. However, each result has different levels of bacterial biological classification's eight taxonomic ranks. Please explain clearly these results.

2. How about the alpha diversity in each group?

3. The wide variety of results makes it a bit difficult to comprehend. In particular, please summarize clearly the differences in intestinal bacteria by colon cancer location, the involvement of obesity, and the usefulness of fecal matter. How about if it could be summarized in a figure or scheme?

4. Did the microbiome data post on public database, such as DDBJ?

6. PLOS authors have the option to publish the peer review history of their article (what does this mean?). If published, this will include your full peer review and any attached files.

Reviewer #1: No

Reviewer #2: No

---

## [Author Response · Author response to Decision Letter 0]

18 May 2023

With this letter we send you the revised version of our manuscript entitled: “Gut Microbiota profiles in feces and paired tumor and non-tumor tissues from Colorectal Cancer patients. Relationship to the Body Mass Index”, to be considered for publication in Plos One, if it is found suitable.

As the Editor suggested, we have revised the analyses in the manuscript after remove the non-CRC group and combine the retrospective and prospective cohorts. Results from the new analyses have been included in the revised version of the manuscript and in the new Figures and Table.

Together with this rebuttal letter, we send a marked-up copy of our manuscript that highlights changes made to the original version (File: Revised Manuscript with Track Changes). Also, an unmarked version of our revised paper without tracked changes has been included as a separate file (Manuscript).

Additional requirements and response to the academic editor and reviewers:

- The role the funders took in the study has to be added in the Funding Statement section of the online submission form: The funders had no role in study design, data collection and analysis, decision to publish, or preparation of the manuscript. Also, the Funding Statement paragraph has been removed from the manuscript.

If the manuscript is accepted for publication in Plos One, the following sentence in its entirety must be included: This study was supported by grant PI19/00073 from the Instituto de Salud Carlos III (Ministerio de Economía y Competitividad), Spain and co-funded by the European Union through the European Regional Development Fund (ERDF) ‘A way to make Europe’. The funders had no role in study design, data collection and analysis, decision to publish, or preparation of the manuscript.

- With respect to “Data Availability”, it should be indicated the following statement: All relevant data are within the manuscript and its Supporting Information files.

Four Supporting Information files in TIF format (S1, S2, S3, S4) have now been included: unmarked revised version, lines 375-394; and files S1 Fig, S2 Fig, S3 Fig and S4 Fig.

- The full name of the ethics committee who approved our study (written consent) has been specified in the “Methods” section of the revised manuscript (lines 96-98 in the unmarked revised version).

- Response to the Reviewer≠2 questions:

1. In the revised version of the manuscript, the CRC cases (retrospectives and prospectives) have been combined, and the control group has been eliminated. In this way, the number of LDA analyses and the number of Figures in the paper has been considerably reduced. The results included in the revised version of the manuscript only consider 3 of the 8 taxonomic ranks from the bacterial biological classification: phylum, order and genus, showing only the most outstanding differences between the study groups considered.

2. The alpha diversity results have been included in the revised version of the manuscript, such as the Reviewer 2 suggested. Specifically, in the Results section (unmarked revised version): lines 170-177, 194-198, 228-230, and 265-266. Complete alpha diversity data and Figures are included as Supporting Information Files (S1 Fig, S2 Fig, S3 Fig, S4 Fig).

3. The current version of the manuscript is more easily understandable. In the conclusions section, we have highlighted the two most relevant aspects, related to the differences in terms of tumor location and the BMI values of the patients (lines 368-372 in the unmarked revised version).

4. This question has been previously answered in this letter: Due to restrictions on data sharing to respect the privacy of participants, the data sets used and/or analysed during the current study will be made available from the corresponding author upon reasonable request. 

All authors have agreed to this submission and have participated in the study to a sufficient extent. We confirm that this is an original study presenting novel work, that it has not been previously submitted to or accepted by any other journal, that is has been approved by all authors, that ethics approval and written informed consent have been obtained, and none author has a conflict of interest. We look forward to receiving your notices regarding the final Editorial decision on our manuscript.

---

## [Decision Letter · Decision Letter 1]

20 Jun 2023

PONE-D-23-02427R1Gut microbiota profiles in feces and paired tumor and non-tumor tissues from Colorectal Cancer patients. Relationship to the Body Mass IndexPLOS ONE

Dear Dr. Iniesta,

Thank you for submitting your manuscript to PLOS ONE. After careful consideration, we feel that it has merit but does not fully meet PLOS ONE’s publication criteria as it currently stands. Therefore, we invite you to submit a revised version of the manuscript that addresses the points raised during the review process.

The revised manuscript is improved but the reviewers still have significant concerns. In particular, both had difficulty identifying the hypotheses of the study, following the Results section, and understanding the conclusions that could be reached from the study. 1) Please revise the manuscript to clarify the hypotheses and provide rationales for each subanalysis shown (e.g. tissue vs. feces, unaffected vs. affected, left vs. right, etc.). In particular, more detail should be provided to justify the comparison of obese vs. non-obese CRC. 2) Please edit the Results and Figures to make it easier to follow the analytic questions and analyses presented. Reviewer #1 gave some suggestions which would help with this. It may be helpful to incorporate alpha diversity results in the supplementary figures into the main figures as Reviewer #2 had difficulty finding the alpha diversity results. 3) The Discussion should address the limitations of the study, including but not limited to the small number of subjects (particularly when comparing subsets of CRC such as obese vs. non-obese) and the lack of statistical adjustment for multiple testing as mentioned by Reviewer #1. 4) The manuscript requires a Data Availability Statement. The standard for data availability for microbiome studies is to deposit deidentified raw sequence data in a public repository. If this is not possible, PLOS One requires the Data Availability Statement to address the following points (https://journals.plos.org/plosone/s/data-availability): “If there are ethical or legal restrictions on sharing a sensitive data set, authors should provide the following information within their Data Availability Statement upon submission: • Explain the restrictions in detail (e.g., data contain potentially identifying or sensitive patient information) • Provide contact information for a data access committee, ethics committee, or other institutional body to which data requests may be sent”

We look forward to receiving your revised manuscript.

Kind regards,

Jonathan Jacobs

Academic Editor

PLOS ONE

Additional Editor Comments:

The revised manuscript is improved but the reviewers still have significant concerns. In particular, both had difficulty identifying the hypotheses of the study, following the Results section, and understanding the conclusions that could be reached from the study.

1) Please revise the manuscript to clarify the hypotheses and provide rationales for each subanalysis shown (e.g. tissue vs. feces, unaffected vs. affected, left vs. right, etc.). In particular, more detail should be provided to justify the comparison of obese vs. non-obese CRC.

2) Please edit the Results and Figures to make it easier to follow the analytic questions and analyses presented. Reviewer #1 gave some suggestions which would help with this. It may be helpful to incorporate alpha diversity results in the supplementary figures into the main figures as Reviewer #2 had difficulty finding the alpha diversity results.

3) The Discussion should address the limitations of the study, including but not limited to the small number of subjects (particularly when comparing subsets of CRC such as obese vs. non-obese) and the lack of statistical adjustment for multiple testing as mentioned by Reviewer #1.

4) The manuscript requires a Data Availability Statement. The standard for data availability for microbiome studies is to deposit deidentified raw sequence data in a public repository. If this is not possible, PLOS One requires the Data Availability Statement to address the following points (https://journals.plos.org/plosone/s/data-availability):

“If there are ethical or legal restrictions on sharing a sensitive data set, authors should provide the following information within their Data Availability Statement upon submission:

• Explain the restrictions in detail (e.g., data contain potentially identifying or sensitive patient information)

• Provide contact information for a data access committee, ethics committee, or other institutional body to which data requests may be sent”

Reviewers' comments:

Reviewer's Responses to Questions

**Comments to the Author**

1. If the authors have adequately addressed your comments raised in a previous round of review and you feel that this manuscript is now acceptable for publication, you may indicate that here to bypass the “Comments to the Author” section, enter your conflict of interest statement in the “Confidential to Editor” section, and submit your "Accept" recommendation.

Reviewer #1: (No Response)

Reviewer #2: (No Response)

2. Is the manuscript technically sound, and do the data support the conclusions?

Reviewer #1: Partly

Reviewer #2: No

3. Has the statistical analysis been performed appropriately and rigorously? 

Reviewer #1: No

Reviewer #2: Yes

4. Have the authors made all data underlying the findings in their manuscript fully available?

Reviewer #1: Yes

Reviewer #2: No

5. Is the manuscript presented in an intelligible fashion and written in standard English?

Reviewer #1: Yes

Reviewer #2: No

6. Review Comments to the Author

Reviewer #1: I appreciate that the authors have revised their manuscript based on my most basic concerns (impossibly small healthy control group, confusing prospective/retrospective case groups). However, the problem remains that they are making far too many comparisons within n=45 CRC cases. Consider that they have [tumor(T)/non-tumor(NT)/feces]*(right/left/rectum)*(obese/non-obese), as well as alpha diversity, beta diversity, and taxonomic comparisons for each of these many, small subgroups.

Please consider these constructive suggestions:

L70 - Remove CRC case-control implication, as there are no non-cancer subjects.

Start a new paragraph for the several parts of the primary results (NT/feces alpha; beta; T/feces alpha; beta; 3-way; T/NT; location).

Add an informative header for each of the above paragraphs (L 168, 178, 199, 204, 220, 227).

In Discussion of right (proximal) vs left (distal), please consider this article:

Proc Natl Acad Sci U S A. 2014 Dec 23; 111(51): 18321–18326.

Published online 2014 Dec 8. doi: 10.1073/pnas.1406199111

In Discussion of limitations, mention lack of statistical adjustment for the many comparisions.

Reviewer #2: This paper by Sofía Tesolato, et al. is very interesting and important paper regarding to gut microbiota profiles in feces and paired tumor and non-tumor tissues from Colorectal Cancer patients. The methods and results are sound and include important evidence.

However, these are major points that should be addressed.

Major points.

1. This include lots of complicated data in the paper, however these results are not well organized. Please summarize the results more understandingly and interpret the data carefully and finally summarize the important points the authors want to emphasize.

2. There are many LDA analysis in the results. However, each result has different levels of bacterial biological classification's eight taxonomic ranks. Please explain clearly these results.

3. How about the alpha diversity in each group?

4. Please mention the public database that you posted, such as DDBJ?

7. PLOS authors have the option to publish the peer review history of their article (what does this mean?). If published, this will include your full peer review and any attached files.

Reviewer #1: No

Reviewer #2: No

---

## [Author Response · Author response to Decision Letter 1]

10 Jul 2023

Response to the Academic Editor:

- As the Editor suggested, we have revised the manuscript to clarify the hypotheses and provide rationales for each subanalysis shown (lines 72-86 and 88-90, in the Introduction section; and lines 305-307, in the Results section of the revised manuscript).

- As the Editor and Reviewer≠1 suggested, we have now improved the presentation of the Results and Figures to make it easier to follow. In addition, alpha diversity Figures have been incorporated into the main Figures, and have been removed from the supplementary material.

- At the end of the Discussion section (lines 431-434, we have included a paragraph addressing the limitations of our study. In this paragraph we consider, in addition to the heterogeneity in age and sex of the populations under study (comment included in the Conclusions section in the previous version of this paper), the limitations due to the number of subjects and the lack of statistical adjustment for multiple testing as mentioned by Reviewer≠1.

- Data Availability Statement has been included. We have deposit the raw sequence data in a public repository:

Submission ID: SUB13573780

BioProject ID: PRJNA989099

http://www.ncbi.nlm.nih.gov/bioproject/989099

Response to the Reviewer≠1 questions:

- CRC case-control implication has been removed (L70 in the previous version of the paper).

- A new paragraph has been started for the several parts of the primary results, adding informative headers, such as Reviewer suggested: lines 186-187 (NT/feces alpha diversity); lines 201-202 (NT/feces beta diversity); lines 219-220 (T/feces alpha diversity); lines 230-231 (T/feces beta diversity); lines 253-254 (T/NT alpha and beta diversity); lines 262-263 (tumor location, alpha and beta diversity); and lines 303-304 (alpha and beta diversity considering BMI values).

- In the Discussion section, the article recommended by Reviewer≠1 (doi: 10.1073/pnas.1406199111) has been considered: lines 396-402 and Reference number 24, in the revised version of the paper.

- At the end of the Discussion section (lines 430-433, we have included a paragraph addressing the limitations of our study. In this paragraph we consider, in addition to the heterogeneity in age and sex of the populations under study (comment included in the Conclusions section in the previous version of this paper), the limitations due to the number of subjects and the lack of statistical adjustment for multiple testing as mentioned by Reviewer≠1.

Response to the Reviewer≠2 questions:

- In the revised version of the manuscript, we have included a paragraph at the end of the Results section (lines 334-347) in which we summarize the most important data from our work. In addition, the Results section has been improved according to the recommendations of the Reviewer≠1. Specifically, a new paragraph has been started for the several parts of the primary results, adding informative headers. Finally, in the Conclusions section we have emphasized the important points of the manuscript.

- At the beginning of the Results section (lines 174-185), as Reviewer≠2 suggested, we have now better explained the results obtained from multiple LDA analyses with different levels of bacterial biological classification's. In order to facilitate the understanding of these data, we clarify that the level of LDA analysis that we showed depended on the number of taxonomic differences present in each particular comparison. Even though we are aware that the differences in bacterial genres may be more specific, when the differences were more abundant (e.g. in the comparison between feces and tumor or non-tumor tissue), we found more interesting to focus on the differences at the higher taxonomic levels (phyla, orders...) as they were more manageable and less confusing than the ones at lower taxonomic levels (LDA at the genus level were very big and difficult to include in a figure). However, when the differences were slighter (e.g. between tumor and non-tumor colorectal tissues), these did not reach the phyla or order levels, so in this case we included the LDA at the genus level with the purpose of refining the comparison.

- Alpha diversity Figures have been incorporated into the main Figures, and have been removed from the supplementary material.

- Data Availability Statement has been included. We have deposited the raw sequence data in a public repository:

Submission ID: SUB13573780

BioProject ID: PRJNA989099

http://www.ncbi.nlm.nih.gov/bioproject/989099

---

## [Decision Letter · Decision Letter 2]

15 Aug 2023

PONE-D-23-02427R2Gut microbiota profiles in feces and paired tumor and non-tumor tissues from Colorectal Cancer patients. Relationship to the Body Mass IndexPLOS ONE

Dear Dr. Iniesta,

Thank you for submitting your manuscript to PLOS ONE. After careful consideration, we feel that it has merit but does not fully meet PLOS ONE’s publication criteria as it currently stands. Therefore, we invite you to submit a revised version of the manuscript that addresses the points raised during the review process.

Please address the three comments from Reviewer #1. Please address the comment from Reviewer #2 on the obesity/CRC relationship by providing further discussion of the potential implications in CRC of the observed differences in microbial composition between obese and normal weight CRC patients in this study.

Also, I have the following comments/questions:

-    “As quality control, a sampling depth with a minimum number of 100k OTUS per sample was chosen.” (line 172) Please clarify if you meant that all samples were rarefied to 100K reads/sample prior to analysis? Also, what was the sequence depth of this study (e.g. mean or median #reads/sample with range)?

-    What preprocessing steps were performed in QIIME2? E.g. de novo OTU picking? What database was used for taxonomy assignment?

-    Was there any difference between left colon and rectum tumors? If so, please add this to Fig. 7. If not, please mention this in the Results.

-    Could you clarify if the analyses shown in Fig. 8 and 9 were done on non-tumor colorectal tissue? If so, was there also a difference by weight category in tumor tissues?

-    Please rearrange the order of columns in Fig. 8 so that overweight is between normal and obese.

-    Did overweight differ from normal weight in beta diversity or LEFSE analyses? If not, please note this in the Results.

-    Why was ANOSIM used in Fig. 10 but PERMANOVA in the other analyses?

-    Please provide a reviewer access link to the Bioproject for this study so that it can be verified.

We look forward to receiving your revised manuscript.

Kind regards,

Jonathan Jacobs

Academic Editor

PLOS ONE

Journal Requirements:

Additional Editor Comments:

Please address the three comments from Reviewer #1. Please address the comment from Reviewer #2 on the obesity/CRC relationship by providing further discussion of the potential implications in CRC of the observed differences in microbial composition between obese and normal weight CRC patients in this study.

Also, I have the following comments/questions:

- “As quality control, a sampling depth with a minimum number of 100k OTUS per sample was chosen.” (line 172) Please clarify if you meant that all samples were rarefied to 100K reads/sample prior to analysis? Also, what was the sequence depth of this study (e.g. mean or median #reads/sample with range)?

- What preprocessing steps were performed in QIIME2? E.g. de novo OTU picking? What database was used for taxonomy assignment?

- Was there any difference between left colon and rectum tumors? If so, please add this to Fig. 7. If not, please mention this in the Results.

- Could you clarify if the analyses shown in Fig. 8 and 9 were done on non-tumor colorectal tissue? If so, was there also a difference by weight category in tumor tissues?

- Please rearrange the order of columns in Fig. 8 so that overweight is between normal and obese.

- Did overweight differ from normal weight in beta diversity or LEFSE analyses? If not, please note this in the Results.

- Why was ANOSIM used in Fig. 10 but PERMANOVA in the other analyses?

- Please provide a reviewer access link to the Bioproject for this study so that it can be verified.

Reviewers' comments:

Reviewer's Responses to Questions

**Comments to the Author**

1. If the authors have adequately addressed your comments raised in a previous round of review and you feel that this manuscript is now acceptable for publication, you may indicate that here to bypass the “Comments to the Author” section, enter your conflict of interest statement in the “Confidential to Editor” section, and submit your "Accept" recommendation.

Reviewer #1: (No Response)

Reviewer #2: All comments have been addressed

2. Is the manuscript technically sound, and do the data support the conclusions?

Reviewer #1: Yes

Reviewer #2: Yes

3. Has the statistical analysis been performed appropriately and rigorously? 

Reviewer #1: Yes

Reviewer #2: Yes

4. Have the authors made all data underlying the findings in their manuscript fully available?

Reviewer #1: Yes

Reviewer #2: Yes

5. Is the manuscript presented in an intelligible fashion and written in standard English?

Reviewer #1: Yes

Reviewer #2: Yes

6. Review Comments to the Author

Reviewer #1: I thank the authors for incorporating my previous suggestions. The manuscript is now in good shape, except for the following very minor considerations:

L174-185: These are not Results. Please drop L174, then move L175-185 to follow the LDA Methods para (L166-172).

L235-236: Why is this sentence underlined?

L255-256: This sentence probably should be: "We investigated possible microbiota differences between tumor and non-tumor tissues (Fig 5). No significant ...."

Reviewer #2: The revised paper by Sofía Tesolato, et al. is very interesting and important paper regarding to gut microbiota profiles in feces and paired tumor and non-tumor tissues from Colorectal Cancer patients. However, the paper improved marginally and lack theimportant evidence that support your hypothesis.

Major points.

This paper includes lots of complicated data in the paper and still complicated. Please summarize the results more understandingly and interpret the data carefully and finally summarize the important points the authors want to emphasize. In addition, you should mention the interpretaion of your data and hypothesiss of the mechanism.

The relation between carcinogenesis and microbiota is unclear. Also, how obesity is related the colon cancer?

In order to clarify these questions, authors should totally rearrange the paper construction. Also functional analysis will be needed if possible.

7. PLOS authors have the option to publish the peer review history of their article (what does this mean?). If published, this will include your full peer review and any attached files.

Reviewer #1: No

Reviewer #2: No

---

## [Author Response · Author response to Decision Letter 2]

12 Sep 2023

Response to the Reviewer≠1 questions:

- Lines 174-185 from the previous version of the manuscript have been rearranged as Reviewer suggested (lines 175-185 in the revised version).

- In the previous version of the manuscript, there was a typo since the sentence indicated by the Reviewer should not have been underlined. This typo has now been corrected (lines 242-243 in the revised version).

- The sentence has been modified according to the Reviewer’s suggestion (lines 262-263 in the revised version).

Response to the Reviewer≠2 questions:

- In the revised version of the manuscript, we have included new comments to clarify the relationship between obesity and CRC. Moreover, we have further discussed the potential implications in CRC of the observed differences in microbial composition between obese and normal with CRC patients in our study (lines 452-456, 466-469, 472-484; and references 29, 30, 34, 35, 37, 38 39, in the revised version of the manuscript).

We consider that the modifications included in the Discussion section of the revised version of the manuscript contribute to better understanding the results of the work that are summarized in lines 368-381. 

It is evident that it would be very interesting to carry out functional analyses in order to reach a complete understanding of the mechanisms involved in the results of the work. However, at this time it is not possible for us to do so. We continue working in order to obtain additional grants that would allow us to carry out functional studies in the future.

Responses to the academic editor comments:

- As the Editor suggested, we have revised the manuscript to clarify the “Statistical analysis” (Materials and methods):

1. Regarding the sequencing depth filter (there is a typo, not being OTUs but reads), that filter was introduced on the first analysis (1st version of the manuscript) that we performed on this dataset. But on the final analysis involving the different cohorts that criteria was not applied, the sequencing depth distribution has a median frequency of 234.647 ranging [27.182-448.128] reads per sample (lines 190-191 of the manuscript).

2. The pre-processing steps that were performed in QIIME2 have been now indicated (lines 157-165 of the manuscript).

- Differences between left colon and rectum tumors were not significant in any of the beta diversity tests considered in this work, and only slight differences between both locations, at the taxonomic level, were detected (lines 310-318 of the manuscript).

- The analyses shown in Fig. 8 and 9 were performed jointly in all colorectal tissues included in the present study (tumor and non-tumor tissues), since no significant differences had been found in any of the alpha and beta diversity tests between both types of tissues (lines 262-263 and 323-326 of the manuscript).

- The order of columns in Fig. 8 has been rearranged, such as Academic Editor suggested.

- Differences in beta diversity and LEfSe analyses between overweight and normal weight have been detailed in this revised version of the manuscript (lines 346-353 and 362-367).

- As it is indicated in lines 354-358 of the manuscript, in relation to the results from the Fig. 10, we only detected bordering statistical significance for the Jaccard index in ANOSIM beta diversity test, without statistical differences in the other analyses.

- We have deposited the raw sequence data in a public repository:

Submission ID: SUB13573780

BioProject ID: PRJNA989099

The access link to the Bioproject (starting August 30, 2023) is:

http://www.ncbi.nlm.nih.gov/bioproject/989099

The revised version of our manuscript now includes changes in the reference list. Specifically, the additional comments in the Discussion section suggested by the Reviewer≠2 have made necessary to include seven new references (29, 30, 34, 35, 37, 38, 39).

---

## [Editor Report · Decision Letter 3]

25 Sep 2023

Gut microbiota profiles in feces and paired tumor and non-tumor tissues from Colorectal Cancer patients. Relationship to the Body Mass Index

PONE-D-23-02427R3

Dear Dr. Iniesta,

We’re pleased to inform you that your manuscript has been judged scientifically suitable for publication and will be formally accepted for publication once it meets all outstanding technical requirements.

Kind regards,

Jonathan Jacobs

Academic Editor

PLOS ONE
---

## [Editor Report · Acceptance letter]

28 Sep 2023

PONE-D-23-02427R3 

Gut microbiota profiles in feces and paired tumor and non-tumor tissues from Colorectal Cancer patients. Relationship to the Body Mass Index. 

Dear Dr. Iniesta:

I'm pleased to inform you that your manuscript has been deemed suitable for publication in PLOS ONE. Congratulations! Your manuscript is now with our production department. 

Kind regards, 

on behalf of

Dr. Jonathan Jacobs 

Academic Editor

PLOS ONE